# Improved Last-iterate Convergence Properties for the FLBR Dynamics

## Abstract

The recent years have seen a surge of interest in algorithms with last-iterate convergence for 2-player games, motivated in part by applications in machine learning. Driven by this, we revisit a variant of Multiplicative Weights Update (MWU), defined recently by Fasoulakis et al. (2022), and denoted as Forward Looking Best Response MWU (FLBR-MWU). These dynamics are based on the approach of extra-gradient methods, with the tweak of using a different learning rate in the intermediate step. So far, it has been proved that this algorithm attains asymptotic convergence but no explicit rate has been known. We answer the open question from Fasoulakis et al. by establishing a geometric convergence rate for the duality gap. In particular, we first show such a rate, of the form $O(c^t)$, until we reach an approximate Nash equilibrium, where $c < 1$ is independent of the game parameters. We then prove that from that point onwards, the duality gap keeps getting decreased with a geometric rate, albeit with a dependence on the maximum eigenvalue of the Jacobian matrix. Finally, we complement our theoretical analysis with an experimental comparison to OGDA, which ranks among the best last-iterate methods for solving 0-sum games. Although in practice it does not generally outperform OGDA, it is often comparable, with a similar average performance.

## 1 Introduction

Our work focuses on learning algorithms with convergence guarantees in 2-player bilinear zero-sum games. This is by now an extensively studied domain, spanning a few decades of research progress already. Given a game described by its payoff matrix, what we are after here is algorithms that eventually reach a Nash equilibrium, from which no player has an incentive to deviate. Some of the earlier and standard results in this area concern convergence *on average*. I.e., it has long been known that by using no-regret algorithms, the empirical average of the players' strategies over time converges to a Nash equilibrium in zero-sum games and to more relaxed equilibrium notions (coarse correlated equilibria) for general games (Freund & Schapire, 1999).

In recent years, the attention of the relevant community has gradually shifted from convergence on average to the more robust notion of *last-iterate convergence*, a property highly desirable from an application perspective. This means that the strategy profile $(x^t, y^t)$, reached at iteration $t$ of an iterative algorithm, converges to the actual equilibrium as $t \to \infty$. Unfortunately, many of the initially developed methods do not satisfy this property. No-regret algorithms, like the Multiplicative Weights Update (MWU) method, are known to converge only in an average sense. In fact, it was shown in Bailey & Piliouras (2018); Mertikopoulos et al. (2018) that several MWU variants do not satisfy last-iterate convergence.

Motivated by these considerations, the last decade has seen a series of works studying last-iterate convergence. The majority of these works have focused on the fundamental class of zero-sum games. Zero-sum games have played an important role in the development of game theory and optimization, and more recently, there has also been a renewed interest, given their relevance in formulating GANs in deep learning (Goodfellow et al., 2014). The positive results that have been obtained for zero-sum games show that improved variants of Gradient Descent such as the Optimistic Gradient Descent/Ascent method (OGDA), or the Extra-Gradient method (EG) attain last iterate convergence. Several other methods have also been obtained and compared to each other with respect to their con-

vergence rate. Overall, one can say that we now have a much better understanding of the learning dynamics that converge in zero-sum games.

Despite the positive progress, however, several important questions still remain unanswered. First, it is often difficult to have tight bounds in analyzing such learning algorithms. Furthermore, even for bilinear, zero-sum games, the best attainable rate of convergence is not yet fully understood. The currently best rate that is applicable to all such games is $O(1/\sqrt{t})$ in terms of the duality gap (Cai et al., 2022; Gorbunov et al., 2022), where the hidden terms in the $O(\cdot)$ notation depend on the game dimension but not on the payoff matrix. In fact this also holds for the more general class of convex-concave min-max optimization problems. It is conceivable though that better rates could be achieved for bilinear games. The work of Wei et al. (2021) establishes a geometric convergence rate of $O(c^t)$ ($c < 1$) for the OMWU method, discussed further in the sequel, albeit with game-dependent parameters within the $O(\cdot)$ term. It remains an open problem whether a geometric convergence rate can be achieved where the dependence is only on the game dimension.

## 1.1 OUR CONTRIBUTIONS

We focus on bilinear zero-sum games and revisit a promising variant of MWU, defined recently in Fasoulakis et al. (2022) and denoted as Forward Looking Best-Response Multiplicative Updates (FLBR-MWU). The dynamics are based on the approach of extra-gradient methods, with the tweak of using a different and more aggressive learning rate in the intermediate step. Our main contributions can be summarized as follows:

- So far, it was only known that the FLBR algorithm attains asymptotic last-iterate convergence, but without any explicit rate. We answer the open question from Fasoulakis et al. (2022) by establishing concrete rates of convergence. Using the duality gap as our metric, we first show a geometric rate, of the form $O(c^t)$, until we reach an approximate Nash equilibrium, for an appropriate level of approximation. More precisely, the parameter $c$ ($c < 1$) is independent of the entries in the payoff matrix, and dependent only on the dimension.
- For games with a unique Nash equilibrium, we further prove that once we reach an approximate equilibrium, the duality gap keeps getting decreased with a geometric rate, until the exact equilibrium solution, albeit with the caveat that there is a dependence on the Jacobian matrix evaluated at the equilibrium. An analogous result also holds for the OMWU method (Wei et al., 2021), as mentioned earlier, but for the KL divergence, and with a different dependence on the game parameters. We view as advantages of our analysis that it yields a simpler and more intuitive proof compared to Wei et al. (2021), and that it also establishes the fast (non game-dependent) convergence to an approximate equilibrium before going towards the exact solution. Furthermore, our proof highlights connections to a neighboring field, as it utilizes ideas from the analysis of the Arimoto-Blahut algorithm (for computing the Shannon's capacity of a discrete memoryless channel).
- We then investigate further properties of FLBR. We prove that it is not a no-regret algorithm, which was not known before. At the same time, we explore aspects of *forgetfulness*, as introduced recently in Cai et al. (2024). We show that in contrast to OMWU, FLBR seems to exhibit forgetfulness, which serves as an indication for fast performance.
- Finally, we perform an experimental comparison of FLBR against OGDA, which is among the best known methods for solving zero-sum games, and against OMWU. We mostly focus on the comparison against OGDA since OMWU is not as competitive in practice (observed also in other recent works). The results reveal that FLBR is generally competitive with OGDA; while it does not outperform OGDA, it exhibits a similar average performance.

Overall, we believe our work provides a more complete treatment on the power and limitations of the FLBR method for bilinear games.

## 1.2 RELATED WORK

There is a vast literature on solving zero-sum games. Given the connection with linear programming, a variety of algorithms focus on optimization and LP-based methods for zero-sum games. Theoretically, the best guarantees for solving the corresponding linear program can be found in Cohen et al. (2021) and van den Brand et al. (2021). Regarding other methods, Hoda et al. (2010) use

Nesterov's first order smoothing techniques to achieve an $\varepsilon$-equilibrium in $O(1/\varepsilon)$ iterations, with the added benefits of simplicity and rather low computational cost per iteration. Following up on that work, Gilpin et al. (2012) propose an iterated version of Nesterov's smoothing technique, which runs within $O(\frac{\|A\|}{\delta(A)} \cdot \ln(1/\varepsilon))$ iterations. This is a significant improvement, with the caveat that the complexity depends on a condition measure $\delta(A)$, with $A$ being the payoff matrix.

In addition to the above, there has been great interest in designing faster learning algorithms for zero-sum games. Although this direction started several decades ago, e.g. with the fictitious play algorithm (Brown, 1951; Robinson, 1951), it has received significant attention more recently given the relevance to formulating GANs in deep learning (Goodfellow et al., 2014) and also other applications in machine learning. Some of the earlier and standard results in this area concern convergence *on average*. That is, it has been known that by using no-regret algorithms, such as the Multiplicative Weights Update (MWU) methods (Arora et al., 2012), the empirical average of the players' strategies over time converges to a Nash equilibrium in zero-sum games. Similarly, one could also utilize Gradient Descent/Ascent (GDA) algorithms. Several other algorithms for zero-sum games are built within the framework of regret minimization both in theory (Carmon et al., 2019; 2024) and in applications (Farina et al., 2021).

Coming closer to our work, within the last decade, there has also been a great interest in algorithms attaining the more robust notion of *last-iterate convergence*. This means that the strategy profile $(x^t, y^t)$, reached at iteration $t$, converges to the actual equilibrium as $t \to \infty$. Negative results in Bailey & Piliouras (2018) and Mertikopoulos et al. (2018) show that several no-regret algorithms, such as many MWU as well as GDA variants, do not satisfy last-iterate convergence. Instead, they may diverge or enter a limit cycle. Motivated by this, there has been a series of works on obtaining algorithms with provable last iterate convergence. The positive results that have been obtained for zero-sum games show that improved versions of Gradient Descent such as the Extra-Gradient method (Korpelevich, 1976) or the Optimistic Gradient method (Popov, 1980) attain last-iterate convergence. In particular, Daskalakis et al. (2018) and Liang & Stokes (2019) show that the optimistic variant of GDA (referred to as OGDA) converges for zero-sum games. Analogously, OMWU (the optimistic version of MWU) also attains last-iterate convergence, as shown in Daskalakis & Panageas (2019) and further analyzed in Wei et al. (2021). Further approaches with convergence guarantees have also been proposed, such as primal-dual hybrid gradient methods (Lu & Yang, 2023). For the case of constrained bilinear zero-sum games, the best convergence rate for the duality gap achieved so far is by (Cai et al., 2022; Gorbunov et al., 2022), which is $O(1/\sqrt{t})$. We note that better rates are achievable for the case of unconstrained bilinear zero-sum games, as e.g., in Mokhtari et al. (2020), but this is an easier problem than what we focus on here. We also note that for the metric of KL divergence, Wei et al. (2021) provide a geometric rate, which is dependent on game parameters.

The method we analyze here is inspired by the general approach of extra-gradient methods, but with the tweak of using different learning rates in the intermediate and final step of each iteration. The idea of using different rates in these two steps of each iteration has also been successful in other recent works. It has been used in Azizian et al. (2020) for a model that concerns the unconstrained bilinear case. Again for the unconstrained case (but even beyond convex-concave functions), the work of Diakonikolas et al. (2021) showed how the use of different learning rates achieved convergence guarantees for their method (referred to as EG+). These ideas have also been applied successfully in the stochastic setting, under noisy gradient feedback, (Hsieh et al., 2020).

Several of these methods have also been studied beyond bilinear payoff functions or beyond zero-sum games, including (Golowich et al., 2020) and also (Diakonikolas et al., 2021) where positive results are shown for a class of non-convex and non-concave problems. There are also negative results however as e.g., established in Daskalakis et al. (2021). Going beyond min-max problems, the work of Patris & Panageas (2024) obtains last-iterate convergence rates in rank-1 games. Results for richer classes of games are provided in Anagnostides et al. (2022), including potential and constant-sum polymatrix games. The landscape, however, is overall less clear.

Finally, further results have been obtained regarding the design of algorithms with convergence guarantees for extensive form games. Although such games are not within the scope of our work, the techniques could prove useful for our restricted class of normal-form zero-sum games. Some

of the main ideas that have been exploited in this literature concern regularization, see e.g. Sokota et al. (2023); Liu et al. (2023) and negative momentum Fang et al. (2025).

## 2 PRELIMINARIES

We consider 2-player $n \times n$ zero-sum games $(R, -R)$. Without loss of generality, we consider that $R \in (0, 1]^{n \times n}$ is the payoff matrix of the row player, and $-R$ is the payoff matrix of the column player.[1] A mixed strategy is a probability distribution $x = (x_1, \ldots, x_n)^\top$ over the standard simplex $\Delta_n$, where the vector $e_i$, with 1 in the $i$-th index and zero elsewhere, corresponds to the pure strategy $i$. The support of a mixed strategy $x$ is the set of the pure strategies to which $x$ assigns positive mass, i.e. $supp(x) = \{i | x_i > 0\}$.

A strategy profile is a tuple $(x, y)$, where $x$ (resp. $y$) is the strategy of the row (resp. column) player. Given a profile $(x, y)$, the expected payoff of the row (resp. column) player is $x^\top R y$ (resp $-x^\top R y$).

**Definition 1** ($\varepsilon$-Nash equilibrium ($\varepsilon$-NE))**.** *A strategy profile $(x, y)$ is an $\varepsilon$-Nash equilibrium of the game $(R, -R)$, with $R \in [0, 1]^{n \times n}$, for $\varepsilon \in [0, 1]$, if and only if, for any $i, j \in [n]$,*

$$x^\top R y + \varepsilon \geq e_i^\top R y, \text{ and } x^\top R y - \varepsilon \leq x^\top R e_j.$$

By setting $\varepsilon = 0$ we have an exact NE. Next we will define our progress measure.

**Definition 2** (Duality Gap)**.** *For zero-sum games, the duality gap function $V$ is defined as*

$$V(x, y) = \max_i e_i^\top R y - \min_j x^\top R e_j.$$

The duality gap is a central notion in game theory as it captures the combined loss of the players for not employing best responses and hence for deviating from a NE, as seen in the fact below.

**Fact 1.** *A strategy profile $(x^*, y^*)$ is a Nash equilibrium of a zero-sum game if and only if it is a (global) minimum of the function $V(x, y)$. Furthermore, if $V(x, y) \leq \varepsilon$, then $(x, y)$ is an $\varepsilon$-NE.*

Before proceeding with the dynamics, we state a simple lemma that relates the $L_1$ norm with the duality gap function, defering its proof in Appendix A.

**Lemma 1.** *For any $x, y$ it holds that $\max_i e_i^\top R y \leq ||y - y^*||_1 + v$ and $\min_j x^\top R e_j \leq ||x - x^*||_1 + v$, where $v$ is the value of the zero-sum game.*

### 2.1 FLBR-MWU DYNAMICS

Here we restate the Forward Looking Best-Response Dynamics as introduced in Fasoulakis et al. (2022). These dynamics follow an extra-gradient approach to find a Nash Equilibrium. Specifically, each iteration involves an intermediate step that serves as a prediction for the update step. The difference with other extra-gradient-like approaches is that different learning rates are used in the intermediate and the final step, which appears crucial to the effectiveness of this approach.

Given an initial strategy profile $(x^0, y^0)$, the two steps of the dynamics can be described as follows:

Step 1 (Intermediate): $\hat{x}_i^t = x_i^{t-1} \cdot \dfrac{e^{\xi \cdot e_i^\top R y^{t-1}}}{\sum_j x_j^{t-1} \cdot e^{\xi \cdot e_j^\top R y^{t-1}}}$, and $\hat{y}_j^t = y_j^{t-1} \cdot \dfrac{e^{-\xi \cdot e_j^\top R^\top x^{t-1}}}{\sum_i y_i^{t-1} \cdot e^{-\xi \cdot e_i^\top R^\top x^{t-1}}}$,

Step 2 (Update): $x_i^t = x_i^{t-1} \cdot \dfrac{e^{\eta \cdot e_i^\top R \hat{y}^t}}{\sum_j x_j^{t-1} \cdot e^{\eta \cdot e_j^\top R \hat{y}^t}}$, and $y_j^t = y_j^{t-1} \cdot \dfrac{e^{-\eta \cdot e_j^\top R^\top \hat{x}^t}}{\sum_i y_i^{t-1} \cdot e^{-\eta \cdot e_i^\top R^\top \hat{x}^t}}$,

When $\xi = \eta$ in the above steps, this is referred to as Mirror-Prox in Nemirovski (2004). Contrary to the conventional wisdom of using rather small learning rates to ensure contraction, our approach utilizes a large value for $\xi$ (aggressive rate for the intermediate exploration step) coupled with a small (conservative) learning rate $\eta \in (0, 1)$ for the update step. Finally, we state an important property that we will use at various points in the sequel:

---

[1] Any game can be transformed to a game with entries in the interval $(0, 1]$ with the same Nash equilibria.

**Lemma 2** (Fasoulakis et al. (2022)). *For any $t > 0$, it holds that as $\xi \to \infty$, $\hat{x}^t$ (resp. $\hat{y}^t$) converges to a best response strategy against $y^{t-1}$ (resp. against $x^{t-1}$).*

**Assumption 1.** We will start the dynamics from the fully uniform distribution, i.e., $x^0 = y^0 = (1/n, \ldots, 1/n)$. Furthermore, we will use a fixed $\eta$, independent of $t$ in all iterations.

## 3 CONVERGENCE ANALYSIS

In this section, we use the duality gap as a metric to study the rate of convergence for FLBR-MWU. This answers the question left open by Fasoulakis et al. (2022). Our analysis consists of two main parts. First, we obtain a geometric rate of convergence until an appropriate approximate equilibrium is reached, where the degree of approximation depends on $\eta$. Then, we show that if $\eta$ is sufficiently small, so as to guarantee that we are close to the exact solution, we can maintain a geometric rate to the exact equilibrium, at the cost of introducing a dependency on the game parameters.

### 3.1 CONVERGENCE TO AN APPROXIMATE EQUILIBRIUM

Let $(x^*, y^*)$ be an arbitrary exact Nash equilibrium, and let $(x^t, y^t)$ be the strategy profile produced by the dynamics at the end of time step $t$. We stress that for the convergence to an approximate equilibrium, we do not need to assume uniqueness.

In our analysis, we utilize the *Kullback-Leibler (KL)* divergence of a profile from $(x^*, y^*)$, defined as follows.

$$D_{KL}((x^*, y^*)||(x^t, y^t)) = \sum\nolimits_{i=1}^{n} x_i^* \cdot \ln(x_i^*/x_i^t) + \sum\nolimits_{j=1}^{n} y_j^* \cdot \ln(y_j^*/y_j^t).$$

Note that by the definition of the dynamics, $x_i^t$ and $y_j^t$ are always positive for any $i, j$ and $t$; hence the ratios above are well-defined. For brevity, we write $D_{KL}((x^*, y^*)||(x^t, y^t))$ as $D^t$. The main technical property for the analysis of reaching an approximate equilibrium is the following lemma.

**Lemma 3.** *It holds that for any $t \geq 1$, and any $\eta \leq 1/2$*

$$\eta \cdot \left[(\hat{x}^t)^\top R y^{t-1} - (x^{t-1})^\top R \hat{y}^t\right] \leq D^{t-1} - D^t + 4\eta^2.$$

This lemma is crucial as it provides a way to correlate the duality gap with the KL divergence. In particular, the left hand side of the formula is a proxy quantity for the duality gap, and converges to it should we choose a large enough $\xi$, as established in the following claim.

**Claim 1.** *For any $(x, y) \in \Delta_n \times \Delta_n$, it holds that $\lim_{\xi \to \infty}[(\hat{x})^\top R y - (x)^\top R \hat{y}] = V(x, y)$.*

From this we have the following:

**Corollary 1.** *For any $t \geq 1$, for any $\eta \leq 1/2$, and for sufficiently large $\xi$, it holds that*

$$V(x^{t-1}, y^{t-1}) \leq \frac{D^{t-1} - D^t}{\eta} + 5\eta.$$

All missing proofs are presented in Appendix B. The next theorem is the main result of this section.

**Theorem 1.** *Under Assumption 1, and for sufficiently small $\eta$ and large $\xi$, the rate of convergence for the KL divergence until we reach a $7\eta$-Nash equilibrium is geometric, in the form $O(\ln n \cdot c^t)$, where $c < 1$ is independent of $t$ and dependent on $n$ and $\eta$. Similarly, the convergence rate of the duality gap to reach a $7\eta$-NE is geometric, in the form $O\left(\frac{\ln n}{\eta} \cdot c^t\right)$.*

*Proof.* Since we have not yet reached a $7\eta$-NE, it holds that $V(x^t, y^t) \geq 7\eta$. Plugging this into Corollary 1 gives us, after rearranging the terms:

$$D^t \leq D^{t-1} - 2\eta^2 = D^{t-1}\left(1 - \frac{2\eta^2}{D^{t-1}}\right)$$

Due to Assumption 1 and the fact that the KL divergence only decreases until we reach an approximate equilibrium (Fasoulakis et al. (2022)), we have that $D^{t-1} \leq D^0 \leq 2\ln(n)$. Thus, we deduce

$$D^t \leq D^{t-1}\left(1 - \frac{\eta^2}{\ln(n)}\right).$$

For $\eta \leq \sqrt{\ln(n)}$, we can unroll the above inequality for all time steps up to $t$ to obtain

$$D^t \leq D^{t-1}\Big(1 - \frac{\eta^2}{\ln(n)}\Big)^t \leq 2\ln(n)\Big(1 - \frac{\eta^2}{\ln(n)}\Big)^t.$$

This means that the KL divergence at time $t$ is bounded by $2\ln(n) \cdot c^t$, where $c < 1$ is independent of $t$ and dependent on $\eta$ and $n$. Coming now to the duality gap, we conclude by Corollary 1 that

$$V(x^t, y^t) \leq \frac{D_{KL}^t((x^*, y^*)||(x^t, y^t))}{\eta} + 5\eta \leq \frac{2\ln(n)}{\eta}\Big(1 - \frac{\eta^2}{\ln(n)}\Big)^t + 5\eta. \tag{1}$$

This upper bound combined with $V(x^t, y^t) \geq 7\eta$ implies that for any time step $t$, until we reach an approximate equilibrium, we have that $\eta \leq \frac{\ln(n)}{\eta}\Big(1 - \frac{\eta^2}{\ln(n)}\Big)^t$. By plugging this back into Equation (1), we eventually get:

$$V(x^t, y^t) \leq \frac{7\ln(n)}{\eta}\Big(1 - \frac{\eta^2}{\ln(n)}\Big)^t. \qquad\qquad \square$$

### 3.2 Convergence to an exact equilibrium under uniqueness

We proceed here to analyze the convergence until the method reaches an exact equilibrium. The technique here is based on a spectral analysis, and for this, we will need to further assume that the game has a unique Nash equilibrium $(x^*, y^*)$. This is a rather common assumption in many related works, and we do not view this as a severe restriction, since the set of zero-sum games with non-unique NE has Lebesgue measure equal to zero (Van Damme, 1991).

Let $t_0$ be the time at which we reach the approximate equilibrium described in Section 3.1 and let $(x^{t_0}, y^{t_0})$ be the corresponding strategy profile. By Theorem 1, it can be extracted that $t_0 = O(\ln(\ln(n))/\ln(\eta))$. The first step in the remaining analysis is to establish that this approximate equilibrium can be close to the actual Nash equilibrium. This is ensured if $\eta$ is sufficiently small.

**Corollary 2** (implied by Theorem 3 in Fasoulakis et al. (2022)). *For any $\delta > 0$, and for any $q \geq 1$, there exists a sufficiently small $\eta$, such that $||(x^*, y^*) - (x^{t_0}, y^{t_0})||_q \leq \delta$.*

Using the above, the asymptotic last-iterate convergence of FLBR (but without a rate) was established in Fasoulakis et al. (2022) by proving that the maximum eigenvalue of the Jacobian matrix at $(x^*, y^*)$ is strictly less than 1. In order to obtain a rate of convergence, we give a more refined analysis, based on a technique utilized in Nakagawa et al. (2021) (namely within the proof of their Theorem 5) for a fundamental problem in information theory.[2]

**Theorem 2.** *Let $(R, -R)$ be a zero-sum game with a unique NE $(x^*, y^*)$. For a sufficiently small $\eta$ and large enough $\xi$, such that $\eta\xi < 1$, the rate of convergence of the duality gap to the NE is geometric for the FLBR dynamics, in the form $A/b^t$, where $A$ and $b$ are determined by the norm of the Jacobian matrix evaluated at $(x^*, y^*)$.*

*Proof.* First, we recall some basic facts established in Fasoulakis et al. (2022) that we use here, and for which uniqueness of equilibrium was needed. FLBR can be easily described as a discrete dynamical system, $\varphi(x, y) = (\varphi_1(x, y), \varphi_2(x, y))$, such that $\varphi(x^t, y^t) = (x^{t+1}, y^{t+1})$, and where $\varphi_{1,i}(x, y)$ is the $i$-th coordinate of $\varphi_1(x, y)$ and similarly for $\varphi_{2,i}(x, y)$, for any $i \in [n]$. The Jacobian of this system is a $2n \times 2n$ matrix, determined by the partial derivatives of $\phi$. Furthermore, when there exists a unique NE and $\eta\xi < 1$, Fasoulakis et al. (2022) proved that there exists some $q \geq 1$, such that

$$\lambda_{\max} \leq ||J(x^*, y^*)||_q < 1,$$

where $\lambda_{\max}$ is the maximum eigenvalue of the Jacobian matrix at the profile $(x^*, y^*)$.

For any $t \geq 0$, consider the strategy profile $(x(p), y(p)) = (1 - p) \cdot (x^*, y^*) + p \cdot (x^t, y^t)$, with $p \in (0, 1)$, as a convex combination of the equilibrium and the profile $(x^t, y^t)$. In our proof, we will eventually need to argue about the Jacobian matrix at such convex combinations.

---

[2]In particular, the problem tackled by Nakagawa et al. (2021) was the convergence analysis of the Arimoto-Blahut algorithm for computing the Shannon's capacity of a discrete memoryless channel.

**Lemma 4.** *For $t \geq t_0$: $||(x^{t+1}, y^{t+1}) - (x^*, y^*)||_q \leq ||(x^t, y^t) - (x^*, y^*)||_q \cdot ||J(x(p^t), y(p^t))||_q$.*

With the above lemma and the continuity of the norm, we can now prove by induction the following:

**Lemma 5.** *Given $\varepsilon > 0$, there exists a sufficiently small $\delta > 0$, such that if $||(x^{t_0}, y^{t_0}) - (x^*, y^*)||_q \leq \delta$, then for any $t \geq t_0$. $||J(x(p^t), y(p^t))||_q < ||J(x^*, y^*)||_q + \varepsilon$.*

Fix now a small $\varepsilon > 0$ and let $\lambda = ||J(x^*, y^*)||_q + \varepsilon$ so that $\lambda < 1$. By Lemma 5 and applying repeatedly Lemma 4, we have that, for any $t \geq t_0$, $||(x^t, y^t) - (x^*, y^*)||_q < \lambda^{t-t_0} \cdot ||(x^{t_0}, y^{t_0}) - (x^*, y^*)||_q$. Therefore, given $\varepsilon > 0$, if we pick a sufficiently small $\eta$, we can ensure that there exists a small $\delta > 0$, such that Corollary 2 holds with this $\delta$, i.e., $||(x^{t_0}, y^{t_0}) - (x^*, y^*)||_q < \delta$, and at the same time Lemma 5 holds with the chosen $\varepsilon$ (and again for this $\delta$). By the equivalence of the norms, all these yield that $||(x^t, y^t) - (x^*, y^*)||_1 < K \cdot \delta \cdot \lambda^{t-t_0}$, for some integer $K > 0$ independent of $t$, and dependent on $q$. This directly bounds the $L_1$ distances from the equilibrium strategies and, by applying Lemma 1, we conclude that

$$V(x^t, y^t) \leq 2K \cdot \delta \cdot \lambda^{t-t_0} + v - v = O(K \cdot \delta \cdot \lambda^t). \qquad \square$$

## 4 REGRET AND FORGETFULNESS

In this section, we focus on some previously unexplored aspects of the FLBR method.

### 4.1 REGRET ANALYSIS

First and most importantly, a fundamental question is whether FLBR is a no-regret algorithm, for which we provide a negative answer. So far, in the literature of methods with last-iterate convergence, there exist both no-regret algorithms (such as Optimistic MWU (Daskalakis & Panageas, 2019)) and algorithms with regret (such as Extra-Gradient). We note that the existence of regret by itself is not necessarily a negative indication for an algorithm's performance. For example, OMWU is outperformed by algorithms that have regret, as discussed in Cai et al. (2024).

**Theorem 3.** *FLBR is not a no-regret algorithm when $\xi$ is sufficiently large.*

We provide a proof outline here, and defer the proofs of the lemmas used below to Appendix C. We first restate the FLBR dynamics, so that each iteration is replaced by two steps. We do this so as to explicitly view FLBR within the framework of online learning algorithms with gradient feedback. Hence, in each step, each player observes the payoff of her pure strategies[3] and updates the mixed strategy accordingly. This gives the following formulation for the row player (and analogously for the column player). For technical convenience, we assume the initial profile is indexed as $(x^{-1}, y^{-1})$:

$$x_i^{2t} = x_i^{2t-1} \cdot \frac{e^{\xi \cdot e_i^\top R y^{2t-1}}}{\sum_j x_j^{2t-1} \cdot e^{\xi \cdot e_j^\top R y^{2t-1}}} \text{ and } x_i^{2t+1} = x_i^{2t-1} \cdot \frac{e^{\eta \cdot e_i^\top R y^{2t}}}{\sum_j x_j^{2t-1} \cdot e^{\eta \cdot e_j^\top R y^{2t}}}, \quad t \geq 0. \quad (2)$$

The example that we use for proving the theorem is the simple Matching Pennies game:

$$R = \begin{bmatrix} +1 & -1 \\ -1 & +1 \end{bmatrix}.$$

We use the initialization $x^{-1} = (1 - \delta, \delta)$ and $y^{-1} = (\delta, 1 - \delta)$, for some small $\delta \in (0, 1/2)$. With this at hand, we can break down the proof of Theorem 3 into the lemmata that follow. For simplicity, we will carry out the proof here assuming $\xi \to \infty$. Under this, note that by Lemma 2, $x^0$ is a best response to $y^{-1}$, and hence we get that $x^0 = (0, 1)$. In fact, we can inductively extend this argument.

**Claim 2.** *For any $t \geq 0$, it holds that $x_1^{2t-1} > \frac{1}{2}$ and $y_1^{2t-1} < \frac{1}{2}$.*

Pairing this with Lemma 2, we get that $x^{2t} = (0, 1)$ (as a best response to $y^{2t-1}$, for any $t$) and symmetrically $y^{2t} = (0, 1)$. Now we are in position to explicitly compute $x_1^{2t-1}$.

---

[3] Note that this is precisely the gradient information, since e.g. $\frac{\partial (x^t)^\top R y^t}{\partial x_i} = e_i^\top R y^t$.

**Lemma 6.** *For sufficiently large $\xi$ we get $x_1^{2t+1} = (1 - \delta)[1 - \delta(1 - e^{2\eta(t+1)})]^{-1}$.*

Clearly, we also have $x_2^{2t+1} = 1 - x_1^{2t+1}$. Due to symmetry we obtain that $y_2^{2t+1} = x_1^{2t+1}$; thus, we have obtained a closed form for the dynamics. The proof is then completed by the next lemma.

**Lemma 7.** *For sufficiently small $\delta$ and sufficiently large $\xi$, the regret of the algorithm for the row player against the fixed strategy $x = (0, 1)$, up until time $T$ is $\Omega(T)$.*

### 4.2 Forgetfulness

In a very recent work, Cai et al. (2024) provided further insights on the performance of OMWU and related dynamics, as compared to OGDA. Their work was motivated by Panageas et al. (2023), where analogous intuitions were given for the fictitious play algorithm. In a nutshell, Cai et al. (2024) attributed the cause of relatively slow convergence of OMWU to a notion they term "forgetfulness". Although they did not provide a formal definition, intuitively, a method that is not forgetful allows the produced strategies to get stuck in almost the same profile over many iterations, which slows down convergence. It was shown that this can occur under OMWU, whereas OGDA does not exhibit the same issues.

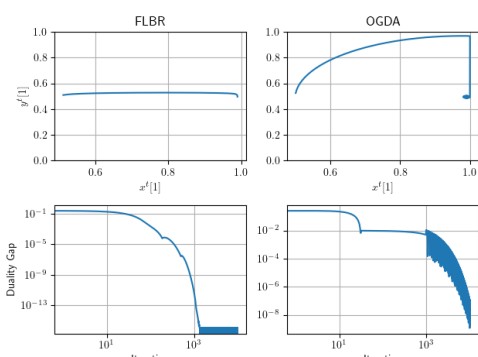

Therefore, the main conclusion of their work is that forgetfulness seems to be a necessary condition for faster performance. Here, we extend their experiment, comparing OGDA and FLBR-MWU. The hard game instance of Cai et al. (2024) for OMWU, parameterized by $\delta \in (0, 1)$, is the following:

$$A_\delta = \begin{bmatrix} \frac{1}{2} + \delta & \frac{1}{2} \\ 0 & 1 \end{bmatrix}$$

The game has a unique equilibrium $(x^*, y^*)$ where $x_1^* = \frac{1}{1+\delta}$ and $y_1^* = \frac{1}{2(1+\delta)}$. In Figure 1, we highlight the behavior of FLBR and OGDA, with $\delta = 10^{-2}$. The upper subfigures show how the first coordinate of $x^t$ and $y^t$ vary over time, starting from the initialization $(x^0, y^0) = (1/2, 1/2)$. In the lower

Figure 1: FLBR vs OGDA in game $A_\delta$.

subfigures, we show the decrease in the duality gap over the iterations. Note that at the equilibrium, $x_1^*$ is close to 1, whereas $y_1^*$ is close to 1/2, and thus close to $y_1^0$. What we observe is that FLBR behaves similarly to OGDA in the sense that it forgets quickly, regarding the coordinate $x_1^t$, and therefore avoiding slowdowns. Furthermore, FLBR does not overshoot $y_1^t$. It increases $y_1^t$ marginally before reaching the actual equilibrium point, whereas OGDA overshoots. This fact justifies the much faster convergence time of FLBR compared to OGDA, as seen in the lower subfigures.

Overall, even though this was only one example, it conveys the intuition that the intermediate step at FLBR, using large $\xi$ has a particular effect in the dynamics: it makes the algorithm forgetful, and thus faster, albeit with the cost of adding regret, as shown in Section 4.1.

## 5 Experimental Evaluation

Experimentally, the method already appeared promising in Fasoulakis et al. (2022). Here, we start by comparing FLBR against OMWU and against OGDA, with the latter being one of the fastest and most well studied last-iterate method for bilinear games (Daskalakis et al., 2018)

We performed three types of comparisons. First, we compare the three methods on random games, and more specifically when the matrices are drawn from a standard Gaussian distribution. Second, we revisit the game $A_\delta$ discussed in Section 4.2. In both experiments, we present one moderately fine-tuned choice of the learning rate $\eta$. Given that OMWU performs quite poorly both in the random games and in $A_\delta$, we then perform further comparisons only between FLBR and OGDA, complemented by more visualizations of different learning rates. Third, to obtain more meaningful comparisons, we sought additional games that are simultaneously far from random and larger in size. To that end, we used the generalized Rock-Paper-Scissors (RPS) game in higher dimensions. In all our experiments, including the additional ones presented in Appendix D, we use a fixed $\xi = 100$ (as a result of our tuning w.r.t. how to set $\xi$).

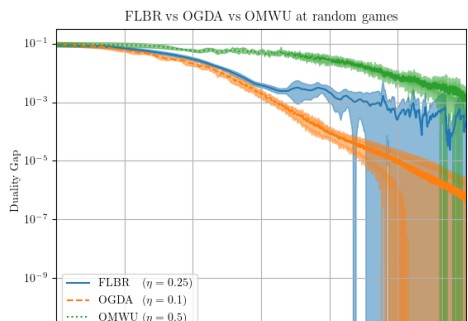
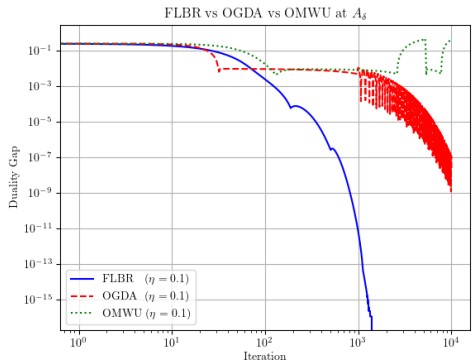

Figure 2: Comparison in Gaussian games     Figure 3: Further comparisons for game $A_\delta$

Our main findings and conclusions are as follows:

- In Figure 2, we see the comparisons on $50 \times 50$ Gaussian random games. The methods are comparable up to a point, with OGDA demonstrating superior performance in both the number of iterations and the time elapsed per game. Nevertheless, FLBR is still close enough and is better than OMWU in time elapsed. The performance of OGDA is explained by Anagnostides & Sandholm (2024), via last iterate analysis under the celebrated framework of *smoothed analysis* (Spielman & Teng, 2004).
- In Figure 3, we see the comparisons for the game $A_\delta$. Here the conclusion reverses: the methods are comparable once again but now FLBR exhibits a clear advantage. OMWU is quite far away.
- In Figures 4 and 5, we see the comparisons for generalized RPS, for dimensions 11 and 101, and for various values of $\eta$. Again, the methods are comparable, with a slight advantage for FLBR.
- Finally, apart from the number of iterations shown in the previous figures, we present some indicative time comparisons between FLBR and OGDA in Tables 1 and 2. Again, the conclusion remains the same, that OGDA performs better in random games, whereas FLBR performs better in RPS, and generally in more structured games (as also verified in our additional experiments in the Appendix).

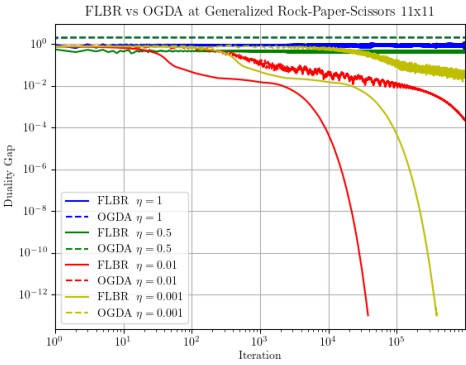
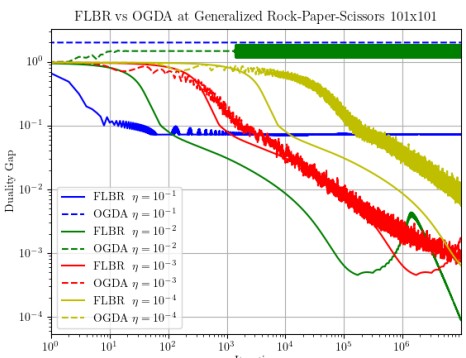

Figure 4: Comparisons over various values of $\eta$     Figure 5: RPS games of higher dimension

Overall, even though the theoretical analysis of FLBR comes with the caveat of game-dependent parameters in its geometric convergence rate, the experiments reveal a competitive performance against OGDA. One more conclusion that arises from the experiments (see Figures 4 and 5) is that FLBR seems to exhibit better robustness to variations in $\eta$, unlike OGDA. We therefore conclude that the combination of different learning rate parameters, $\eta$ and $\xi$, in FLBR can be viewed as promising direction for future work. As a step towards further explorations for the performance of FLBR, it would be interesting to study if our results generalize beyond bilinear payoffs. We have conducted some initial experimentation on this, presented in Section D.3.

Table 1: Comparison in Gaussian games

| | Time (sec) to accuracy | | | |
|---|---|---|---|---|
| | $10^{-2}$ | $10^{-3}$ | $10^{-4}$ | $10^{-5}$ |
| OGDA | 0.005 | 0.026 | 0.155 | 1.72 |
| FLBR | 0.005 | 0.14 | 0.8 | 3.87 |

Table 2: Comparison in RPS

| | Time (sec) to accuracy | | | |
|---|---|---|---|---|
| | $10^{-2}$ | $10^{-3}$ | $10^{-4}$ | $10^{-5}$ |
| OGDA | 4.73 | 14.45 | 24.28 | 34.00 |
| FLBR | 0.08 | 0.11 | 0.15 | 0.22 |

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

## A    MISSING PROOFS FROM SECTION 2

*Proof of Lemma 1.*  We have that for any $i$,

$$|e_i^\top Ry - e_i^\top Ry^*| = \Big| \sum_j R_{ij} \cdot y_j - \sum_j R_{ij} \cdot y_j^* \Big|$$

$$= \Big| \sum_j R_{ij} \cdot (y_j - y_j^*) \Big|$$

$$\leq \sum_j |R_{ij} \cdot (y_j - y_j^*)|$$

$$= \sum_j R_{ij} \cdot |(y_j - y_j^*)|$$

$$\leq \sum_j |(y_j - y_j^*)|$$

$$= ||y - y^*||_1.$$

Thus, if $b = \arg\max_i e_i^\top Ry$, then $\max_i e_i^\top Ry = e_b^\top Ry \leq ||y - y^*||_1 + e_b^\top Ry^* \leq ||y - y^*||_1 + v$. The second part of the lemma follows in a similar manner. □

## B    MISSING PROOFS FROM SECTION 3

### B.1    PROOF OF LEMMA 3

*Proof.*  We first rewrite the KL terms, by using the definition of the dynamics.

$$D_{KL}((x^*, y^*)||(x^{t-1}, y^{t-1})) - D_{KL}((x^*, y^*)||(x^t, y^t))$$

$$= \sum_{i=1}^n x_i^* \cdot \ln(x_i^t / x_i^{t-1}) + \sum_{j=1}^n y_j^* \cdot \ln(y_j^t / y_j^{t-1})$$

$$= \sum_{i=1}^n x_i^* \cdot \ln e^{\eta \cdot e_i^\top R\hat{y}^t} - \ln \Big( \sum_{k=1}^n x_k^{t-1} \cdot e^{\eta \cdot e_k^\top R\hat{y}^t} \Big)$$

$$+ \sum_{j=1}^n y_j^* \cdot \ln e^{-\eta \cdot e_j^\top R^\top \hat{x}^t} - \ln \Big( \sum_{k=1}^n y_k^{t-1} \cdot e^{-\eta \cdot e_k^\top R^\top \hat{x}^t} \Big)$$

$$= \eta \cdot (x^*)^T R\hat{y}^t - \eta \cdot (y^*)^T R^\top \hat{x}^t - \ln \Big( \sum_{k=1}^n x_k^{t-1} \cdot e^{\eta \cdot e_k^\top R\hat{y}^t} \Big) - \ln \Big( \sum_{k=1}^n y_k^{t-1} \cdot e^{-\eta \cdot e_k^\top R^\top \hat{x}^t} \Big).$$

We now use the Taylor expansion of the exponential function in the arguments of the last two logarithms. For the first logarithmic term, this becomes:

$$\ln \Big( \sum_{k=1}^n x_k^{t-1} \cdot e^{\eta \cdot e_k^\top R\hat{y}^t} \Big) = \ln \Big( 1 + \eta \cdot (x^{t-1})^\top R\hat{y}^t + \sum_{k=1}^n x_k^{t-1} \sum_{\ell \geq 2} \frac{(\eta \cdot e_k^\top R\hat{y}^t)^\ell}{\ell!} \Big)$$

$$\leq \ln \Big( 1 + \eta \cdot (x^{t-1})^\top R\hat{y}^t + 2\eta^2 \Big).$$

For the above, we used the fact that $\sum_{\ell \geq 2} \frac{(\eta \cdot e_k^\top R\hat{y}^t)^\ell}{\ell!} \leq \frac{\eta^2}{1-\eta} \leq 2\eta^2$, since $\eta \leq 1/2$. By exploiting now the inequality $\ln(x) \leq x - 1$, we finally obtain the bound

$$\ln \Big( \sum_{k=1}^n x_k^{t-1} \cdot e^{\eta \cdot e_k^\top R\hat{y}^t} \Big) \leq \eta \cdot (x^{t-1})^\top R\hat{y}^t + 2\eta^2.$$

By carrying out similar calculations for the second logarithmic term, we will also get that

$$\ln \Big( \sum_{k=1}^n y_k^{t-1} \cdot e^{-\eta \cdot e_k^\top R^\top \hat{x}^t} \Big) \leq -\eta \cdot (\hat{x}^t)^\top Ry^{t-1} + 2\eta^2.$$

This gives us:

$$D_{KL}((x^*, y^*)||(x^{t-1}, y^{t-1})) - D_{KL}((x^*, y^*)||(x^t, y^t))$$
$$\geq \eta \cdot (x^*)^T R \hat{y}^t - \eta \cdot (y^*)^T R^\top \hat{x}^t - \eta \cdot (x^{t-1})^\top R \hat{y}^t + \eta \cdot (\hat{x}^t)^\top R y^{t-1} - 4\eta^2.$$

By rearranging the terms, we obtain that

$$\eta \cdot \left((\hat{x}^t)^\top R y^{t-1} - (x^{t-1})^\top R \hat{y}^t\right) \leq D_{KL}((x^*, y^*)||(x^{t-1}, y^{t-1})) - D_{KL}((x^*, y^*)||(x^t, y^t)) + 4\eta^2$$
$$- \eta \cdot (x^*)^\top R \hat{y}^t + \eta \cdot (\hat{x}^t)^\top R y^*.$$

Note now that since $(x^*, y^*)$ is a Nash equilibrium, and we are in a zero-sum game, then we know that $(x^*)^\top R \hat{y}^t \geq v$, where $v$ is the value of the game. Similarly, $(\hat{x}^t)^\top R y^* \leq v$. Hence these terms cancel out in the above equation and the proof is complete. □

## B.2 Proofs of Claim 1 and Corollary 1

*Proof.* Recalling Definition 2 we have that $V(x, y) = \max_i e_i^\top R y - \min_j (x)^\top R e_j$. But by Lemma 2, we have that $\hat{x}$ converges to a best response against $y$, and similarly for $\hat{y}$, which completes the proof. □

*Proof.* By Claim 1, we know that $(\hat{x}^\top R y - x^\top R \hat{y}) \to V(x, y)$ as $\xi \to \infty$. From the definition of the limit and the continuity of $\hat{x}, \hat{y}$ as functions of $\xi$ we get that for every $(x, y) \in \Delta_n \times \Delta_n$ and any $\varepsilon > 0$ there exists a $\xi_0(x, y, \varepsilon)$ such that $|(\hat{x}^t)^\top R y^{t-1} - (x^{t-1})^\top R \hat{y}^t - V(x^{t-1}, y^{t-1})| \leq \epsilon$. By setting $\varepsilon = \eta$ and simplifying the $\xi$ notation we deduce that there exists a $\xi(x, y)$ such that

$$V(x, y) \leq \hat{x}^\top R y - x^\top R \hat{y} + \eta, \quad \forall \xi \geq \xi_0(x, y)$$

We select our constant $\xi = \max_{x,y} \xi_0(x, y)$ to enable us to argue for every pair of iterates $(x^t, y^t)$ and via Lemma 3 we get the desired inequality. □

## B.3 Proof of Lemma 4

First we show the following claim that we use in the proof of our Lemma.

**Claim 3.** $\dfrac{d\varphi(x(p), y(p))}{dp} = J(x(p), y(p)) \cdot \left(x^t - x^*, y^t - y^*\right).$

In the equation above, the term $(x^t - x^*, y^t - y^*)$ is a vector of $2n$ coordinates, where for each $i \in [n]$ the $i$-th coordinate equals $x_i^t - x_i^*$, and the $(n + i)$-th coordinate equals $y_i^t - y_i^*$.

*Proof.* For the row player, we have that for any $i$,

$$\frac{d\varphi_{1,i}(x(p), y(p))}{dp} = \sum_k \frac{dx_k(p)}{dp} \cdot \frac{d\varphi_{1,i}(x(p), y(p))}{dx_k(p)} + \sum_\ell \frac{dy_\ell(p)}{dp} \cdot \frac{d\varphi_{1,i}(x(p), y(p))}{dy_\ell(p)}$$
$$= \sum_k \left(x_k^t - x_k^*\right) \cdot J(x(p), y(p))_{ik} + \sum_\ell \left(y_\ell^t - y_\ell^*\right) \cdot J(x(p), y(p))_{i,n+\ell}$$

The above hold because $\frac{dx_k(p)}{dp} = x_k^t - x_k^*$ and $\frac{dy_\ell(p)}{dp} = y_\ell^t - y_\ell^*$. Analogous expressions hold for $\varphi_2$ as well, thus we conclude that

$$\frac{d\varphi(x(p), y(p))}{dp} = J(x(p), y(p)) \cdot \left(x^t - x^*, y^t - y^*\right). \qquad \square$$

*Proof.* By the Mean Value Theorem (applied for our function $f^t = \varphi(x(p), y(p)) : \mathbb{R} \to \mathbb{R}^{2n}$), for each time $t$, there is a $p^t \in (0, 1)$ s.t.

$$
\begin{aligned}
||(x^{t+1}, y^{t+1}) - (x^*, y^*)||_q &= \left|\left|\Big(\varphi_1(x^t, y^t), \varphi_2(x^t, y^t)\Big) - \Big(\varphi_1(x^*, y^*), \varphi_2(x^*, y^*)\Big)\right|\right|_q \\
&= ||f^t(1) - f^t(0)||_q \\
&\leq \left|\left|\frac{df^t(p)}{dp}|_{p=p^t}\right|\right|_q \cdot (1 - 0) \\
&= \left|\left|\Big((x^t, y^t) - (x^*, y^*)\Big) \cdot J(x(p^t), y(p^t))\right|\right|_q \\
&\leq ||(x^t, y^t) - (x^*, y^*)||_q \cdot ||J(x(p^t), y(p^t))||_q
\end{aligned}
$$

where the second inequality holds by the properties of the $q$-norm. $\qquad\square$

### B.4 PROOF OF LEMMA 5

*Proof.* For the basis of the induction, consider $t = t_0$. Regarding the Jacobian, first note that

$$
\begin{aligned}
||(x(p^{t_0}), y(p^{t_0})) - (x^*, y^*)||_q &= ||(1 - p^{t_0})(x^*, y^*) + p^{t_0}(x^{t_0}, y^{t_0}) - (x^*, y^*)||_q \\
&= ||p^{t_0}(x^{t_0}, y^{t_0}) - p^{t_0}(x^*, y^*)||_q \\
&\leq ||(x^{t_0}, y^{t_0}) - (x^*, y^*)||_q
\end{aligned}
$$

Furthermore, by the continuity of the norm, for the given $\varepsilon$, there exists $\delta > 0$ s.t. if $||(x^*, y^*) - (x(p^{t_0}), y(p^{t_0}))||_q < \delta$, then $\Big|||J(x(p^{t_0}), y(p^{t_0}))||_q - ||J(x^*, y^*)||_q\Big| < \varepsilon$. Therefore, if we use this value of $\delta$, we get that if $||(x^{t_0}, y^{t_0}) - (x^*, y^*)||_q \leq \delta$, then $||(x(p^{t_0}), y(p^{t_0})) - (x^*, y^*)||_q < \delta$ (by the previous analysis), and consequently $||J(x(p^{t_0}), y(p^{t_0}))||_q < ||J(x^*, y^*)||_q + \varepsilon$. This establishes the basis.

For the induction step, assume that the condition holds for some $t \geq t_0$. We will establish it for $t + 1$.

Since we have assumed that $\varepsilon$ satisfies $||J(x^*, y^*)||_q + \varepsilon < 1$, the induction hypothesis yields that $||J(x(p^t), y(p^t))||_q < 1$. Using this and Lemma 4, we get that $||(x^{t+1}, y^{t+1}) - (x^*, y^*)||_q < ||(x^t, y^t) - (x^*, y^*)||_q$. This also implies that if $||(x^{t_0}, y^{t_0}) - (x^*, y^*)||_q \leq \delta$, this propagates throughout all the iterations for the same $\delta$, so that $||(x^{t+1}, y^{t+1}) - (x^*, y^*)||_q < \delta$. And this in turn yields

$$
\begin{aligned}
||(x(p^{t+1}), y(p^{t+1})) - (x^*, y^*)||_q &= ||(1 - p^{t+1})(x^*, y^*) + p^{t+1}(x^{t+1}, y^{t+1}) - (x^*, y^*)||_q \\
&= ||p^{t+1}(x^{t+1}, y^{t+1}) - p^{t+1}(x^*, y^*)||_q \\
&\leq ||(x^{t+1}, y^{t+1}) - (x^*, y^*)||_q \\
&< \delta
\end{aligned}
$$

To finish the proof, we use the same argument as in the induction basis. Namely, by the continuity of the norm, for the given $\varepsilon$, and for the $\delta$ that was identified in the induction basis, we will have that $\Big|||J(x(p^{t+1}), y(p^{t+1}))||_q - ||J(x^*, y^*)||_q\Big| < \varepsilon$, and thus

$$
||J(x(p^{t+1}), y(p^{t+1}))||_q < ||J(x^*, y^*)||_q + \varepsilon < 1.
$$

$\qquad\square$

## C   Missing proofs from Section 4

### C.1   Proof of Lemma 6

*Proof.* Recall that

$$x_1^{2t+1} = x_1^{2t-1} \cdot \frac{e^{\eta \cdot e_1^\top R y^{2t}}}{\sum\limits_j x_j^{2t-1} \cdot e^{\eta \cdot e_j^\top R y^{2t}}} = x_1^{2t-1} \cdot \frac{e^{-\eta}}{\sum\limits_j x_j^{2t-1} \cdot e^{\eta \cdot e_j^\top R y^{2t}}}$$

$$x_2^{2t+1} = x_2^{2t-1} \cdot \frac{e^{\eta \cdot e_2^\top R y^{2t}}}{\sum\limits_j x_j^{2t-1} \cdot e^{\eta \cdot e_j^\top R y^{2t}}} = x_2^{2t-1} \cdot \frac{e^{\eta}}{\sum\limits_j x_j^{2t-1} \cdot e^{\eta \cdot e_j^\top R y^{2t}}}$$

For brevity, let $x_1^{2t+1} = a^t$ and $x_2^{2t+1} = b^t$ we get that

$$a^t = a^{t-1} \cdot \frac{e^{-\eta}}{a^{t-1} e^{-\eta} + \beta^{t-1} e^{\eta}}$$

$$b^t = b^{t-1} \cdot \frac{e^{\eta}}{a^{t-1} e^{-\eta} + \beta^{t-1} e^{\eta}}$$

Note that $a^t + b^t = 1$ so we get

$$a^t = a^{t-1} \cdot \frac{e^{-\eta}}{a^{t-1} e^{-\eta} + (1 - a^{t-1}) e^{\eta}} = \frac{a^{t-1} e^{-\eta}}{a^{t-1}(e^{-\eta} - e^{\eta}) + e^{\eta}} \implies$$

$$\frac{1}{a^t} = 1 - e^{2\eta} + e^{2\eta} \frac{1}{a^{t-1}} \implies$$

$$\frac{1}{a^t} - 1 = e^{2\eta} \left( \frac{1}{a^{t-1}} - 1 \right) \implies$$

$$\frac{1}{a^t} - 1 = e^{2\eta(t+1)} \left( \frac{1}{a^{-1}} - 1 \right)$$

Recall that $a^{-1} = x_1^{-1} = 1 - \delta$ so we get that

$$\frac{1}{a^t} = 1 + e^{2\eta(t+1)} \frac{\delta}{1 - \delta} \implies x_1^{2t+1} = \frac{1 - \delta}{1 - \delta(1 - e^{2\eta(t+1)})} \qquad \square$$

### C.2   Proof of Lemma 7

*Proof.* For a given $T$, we compute the total payoff of the row player for the first $2T$ iterations when both players use FLBR. Since at the even steps of this process the strategy of both players is $(0, 1)$, we get:

$$\sum_{i=0}^{2T} x^{i\top} R y^i = T \cdot (0,1)^\top R(0,1) + \sum_{i=0}^{T} x^{2i+1\top} R y^{2i+1}$$

$$= T + \sum_{i=1}^{T} (a^t, 1 - a^t)^\top R(1 - a^t, a^t)$$

$$= T + \sum_{i=1}^{T} (a^t, 1 - a^t)^\top (1 - 2a^t, -1 + 2a^t)$$

$$= T + \sum_{i=1}^{T} a^t - 2(a^t)^2 - 1 + 2a^t + a^t - 2(a^t)^2$$

$$= \sum_{i=1}^{T} 4a^t(1 - a^t)$$

where once again we set $x_1^{2t+1} = a^t$.

Next, we compute the payoff of the fixed strategy $x^* = (0,1)$ for the row player, against the column player playing in each iteration the FLBR strategy $y^i$ as computed by the previous analysis. This is equal to:

$$\sum_{i=0}^{2T} x^{i\top} R y^i = T \cdot (0,1)^\top R(0,1) + \sum_{i=0}^{T} (0,1)^\top R y^{2i+1}$$

$$= T + \sum_{i=0}^{T} (0,1)^\top (1 - 2a^t, -1 + 2a^t)$$

$$= \sum_{i=0}^{T} 2a^t$$

Hence, the regret for the row player when choosing her FLBR strategy against $x^*$ is

$$\text{Reg}_{\text{FLBR}} \geq \sum_{i=0}^{T} 2a^t - \sum_{i=1}^{T} 4a^t(1 - a^t) = \sum_{i=0}^{T} 2a^t(2a^t - 1)$$

To upper bound the expression we use that $a^t = 1/2$ hence we have that

$$\frac{1 - \delta(1 - e^{2\eta(T+1)})}{1 - \delta} = 2$$

$$\delta e^{2\eta(T+1)} = 1 - \delta$$

$$2\eta(T+1) = \ln\left(\frac{1-\delta}{\delta}\right)$$

Thus, up to time $\lceil \frac{T+1}{2} \rceil$ we have that

$$a^t \geq \frac{1 - \delta}{1 - \delta\left(1 - \sqrt{\frac{1-\delta}{\delta}}\right)} = \frac{1 - \delta}{1 - \delta + \sqrt{\delta - \delta^2}}$$

For $\delta \to 0$ the expression tends to 1 so there is a sufficiently small $\delta$ such that $a^t \geq .95$ for $t \leq \lceil \frac{T+1}{2} \rceil$. Piecing everything together we get that

$$\text{Reg}_{\text{FLBR}} \geq \sum_{i=0}^{T} 2a^t(2a^t - 1)$$

$$\geq \sum_{i=0}^{\lceil \frac{T+1}{2} \rceil} 2a^t(2a^t - 1)$$

$$\text{Reg}_{\text{FLBR}} \geq 0.855 \cdot T \quad \text{over } 2T \text{ rounds,}$$

which completes the proof. $\qquad \square$

## D  ADDITIONAL EXPERIMENTS

Our additional experiments follow a similar line of thought as the ones presented in the main paper. Namely, we start with random Gaussian games, where OGDA has a slight advantage over FLBR and then we present constructions of not so random games, with some inherent structure, which slow down OGDA but not FLBR.

**Initializations**   As stated in Assumption 1, for the theoretical part of the paper we always initialize FLBR with the uniform distribution, i.e. $x_i = y_i = 1/n$. Here we deem useful to explore more options. Specifically, we test the following starting points:

- Uniform distribution.
- Almost pure strategy profile: $x_1 = y_1 = 1 - 1/n$ and $x_i = y_i = \frac{1}{n(n-1)}$
- Random: we sample $x, y$ from $U(0,1)$ and then rescale them
- Sequential: $x_i = y_i = \frac{2i}{n(n+1)}$

**Assumptions on $\eta, \xi$** In the theoretical part of the paper, we did not need any major assumption for $\eta$ and $\xi$ (apart from $\xi$ being large enough) for reaching an approximate equilibrium. However, for the convergence to the exact solution, we needed to use $\eta\xi < 1$, to prove Theorem 2. In our experiments, we also tested combinations of values for these two parameters that violate this condition. What we observe experimentally is that the method can perform well even without this constraint (recall e.g., that in the main paper, we also used $\xi = 100$ and values of $\eta$ for which $\eta\xi > 1$), but certainly not for any arbitrary combination.

## D.1 RANDOM GAMES

In addition to the $1000 \times 1000$ Gaussian games presented in the main paper, we see in Figures 6 and 7 the comparisons between FLBR and OGDA for further Gaussian games of dimensions 50 and 500, where each entry of the payoff matrix is filled by sampling from the Gaussian distribution. What we observe is similar to the plots presented also in the main paper for Gaussian games, namely that OGDA performs better (as expected by the existing smoothed analysis for OGDA) and that FLBR is close but on average slower than OGDA.

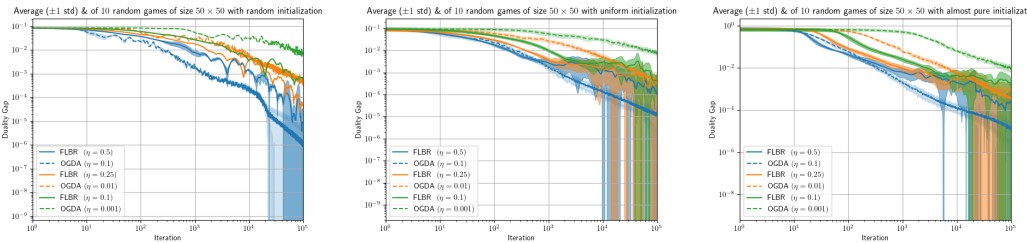

Figure 6: Random Gaussian $50 \times 50$ games with various initializations.

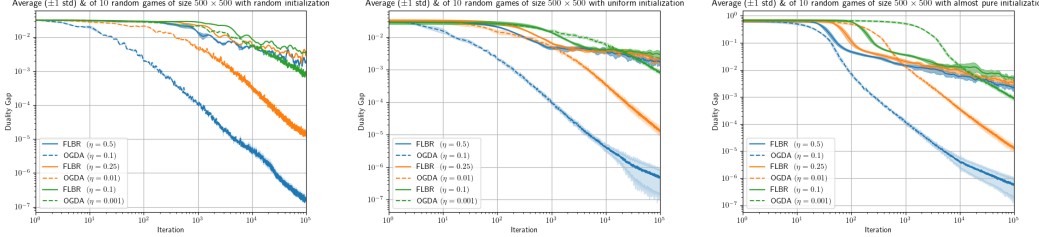

Figure 7: Random Gaussian $500 \times 500$ games with various initializations.

## D.2 STRUCTURED GAMES

We have already presented in the main paper our results on the Generalized Rock-Papers-Scissors game, which is arguably among the most famous zero-sum game. Here we also present comparisons using two more classes of more structured games.

First, we performed comparisons for games where the payoff matrix $R$ is of low rank. Such games differ from random games, where with high probability the matrix has full rank. We constructed matrices, where the rank is approximately 5-10% of the dimension.

Interestingly, what we observe in Figures 8 and 9, is that FLBR is performing better than OGDA. The figures depict the comparisons for $50 \times 50$ games where the rank is 5 and for $500 \times 500$ games

with rank equal to 25. An additional observation is that FLBR seems more robust against the various initializations that were used. For example OGDA, under the random and the uniform initialization does not converge for some choices of $\eta$.

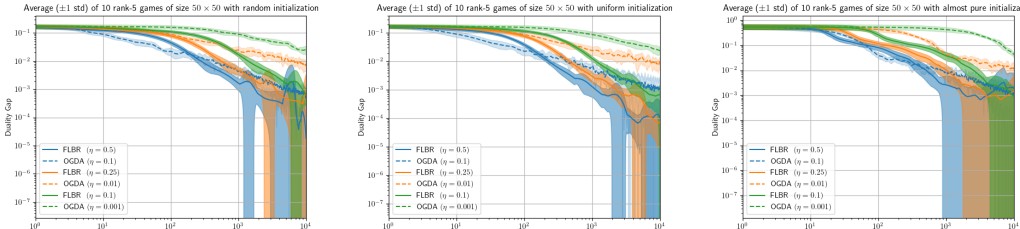

Figure 8: Games with low rank payoff matrix of size $50 \times 50$ with various initializations.

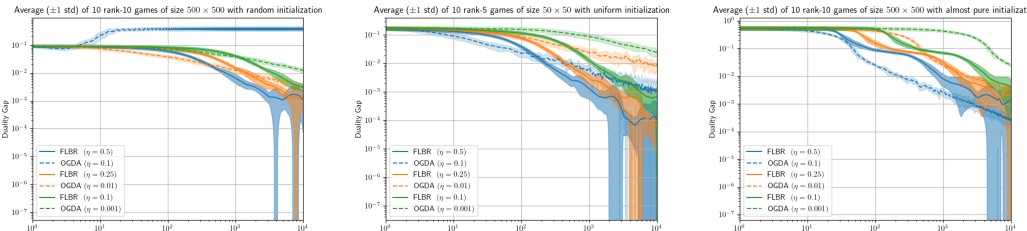

Figure 9: Games with low rank payoff matrix of size $500 \times 500$ with various initializations.

Moving on, we also tested a class of symmetric zero-sum games, which again is more structured than random games. In order to construct such families, we used the following formula for filling in the entries of the payoff matrix, where $P_{ij}^n$ is the entry of $P$ at $(i,j)$ when $P$ is $n \times n$. Here symmetry is enforced, given the dependence on $i + j$.

$$P_{ij}^n = \frac{1}{n}(i + j - 2) \bmod n \tag{3}$$

We note that for this class, we did not use the uniform initialization as this is an equilibrium of the game. What we observe in Figures 10 and 11, is that FLBR is having an advantage over OGDA for smaller dimensions, while OGDA becomes just slightly better, for the sequential and the almost pure initialization. The two methods have a very similar performance under the random initialization. Again, we observe a better robustness of FLBR with respect to the various initializations and the values of $\eta$. For example, we see that OGDA does not manage to converge for some of the choices used for $\eta$.

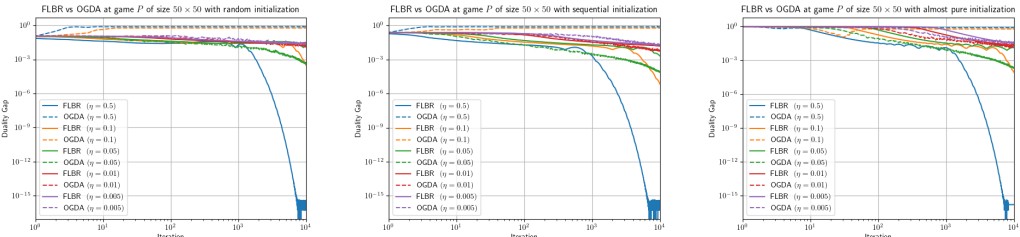

Figure 10: Structured games defined by Equation (3), of size $50 \times 50$ with various initializations.

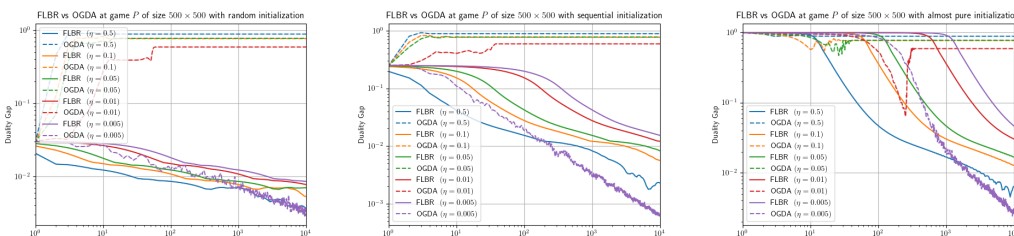

Figure 11: Structured games defined by Equation (3), of size $500 \times 500$ with various initializations.

Overall, a general conclusion that can be extracted from our experiments is that the two methods are of comparable performance, with OGDA doing better for randomly generated games, where FLBR gains an advantage for more structured games.

### D.3 EXPERIMENTATION BEYOND BILINEAR GAMES

Finally, in our last set of experiments, we also tried to investigate if our method is convergent when we move away from bilinear games. To that end, we implemented the method as is for two well studied settings: 1) convex-concave and 2) potential games.

For the first setting, we tested the method on the min-max objective $f(x, y) = \|x - y\|^2 = \sum_{i \in [n]} (x_i - y_i)^2$. The results are shown in Figure 12 for vectors of size 5. The equilibrium here is that both players get a zero payoff, and as we see in Figure 12, FLBR does not manage to converge. This is still far from conclusive, and it remains an interesting direction for future work to investigate under what families of convex-concave functions we could have convergence of FLBR or if the method needs adaptation to extend to more general domains.

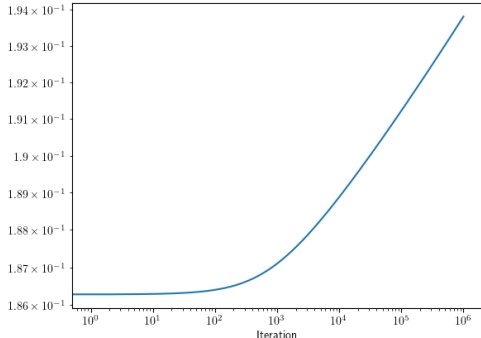

Figure 12: FLBR in a convex-concave setting with the payoff function $\|x - y\|^2$.

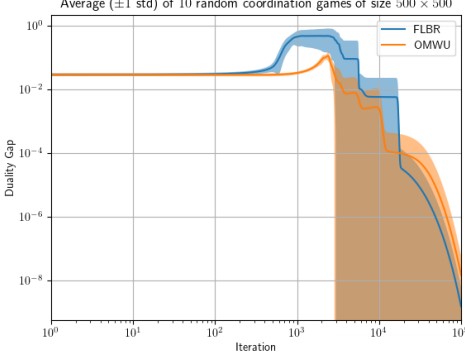

Figure 13: FLBR vs OMWU in random coordination games.

For potential games, we considered the simple scenario of a coordination game where both players have the same payoff matrix. We present the averaged results for 10 games of size $500 \times 500$, drawn from the standard gaussian in Figure 13. For reference, we compare against OMWU. Both methods are executed with stepsize $\eta = 0.1$. We observe that FLBR-MWU does converge and it also has comparable performance, and nearly identical after a certain point, with OMWU. An interesting phenomenon that occurs is that after an initial phase of almost no change, it appears as if FLBR will diverge. But then the behavior of the dynamics change, and we see an alternation between sharp drops in the duality gap and almost constant phases. Understanding this behavior, as well as studying the last iterate rate of FLBR in potential games, is an interesting topic of further research.

