# OpenReview forum: "Improved Last-iterate Convergence Properties for the FLBR Dynamics"
_ICLR.cc/2026/Conference — Submitted to ICLR 2026_

### Official Review · Reviewer_rnbp · 2025-10-22

**Soundness:** 1
**Presentation:** 3
**Contribution:** 2
**Rating:** 2
**Confidence:** 4

**Summary:**

This paper studies the FLBR-MWU dynamics, which is an extragradient version of OMWU, but where the intermediate stepsize is large. The authors show new convergence results for these dynamics, giving a concrete rate for the last-iterate performance.

**Strengths:**

- The OMWU dynamics are an important learning algorithm, and the extragradient variant is of interest as well
- Having a concrete rate on the last iterate is of interest

**Weaknesses:**

1. A missing interpretation in the paper is the following: if we are choosing "large $\xi$" then we are turning the algorithm into a well-known algorithm. In particular, for very large $\xi$, we get that each player is simply best responding to the other player. At that point, you have decoupled the dynamics, and we are simply looking at the intermediate step as a subgradient calculation. At that point, you can instead describe FLBR as follows: we are running subgradient descent, but instead of using projected gradient descent, we use mirror descent with an entropy regularizer. This has implications e.g. for the forgetfulness question. More generally, this interpretation suggests that for large $\xi$, this is not really a "dynamics," it is simply running entropy-based subgradient descent on each player's problem $\min_x f(x), \max_y g(y)$, where $f,g$ encode the fact that the other player best responds. Since the paper repeatedly appeals to "sufficiently large $\xi$" with no upper bound on how large $\xi$ needs to be, I think it is fair to say that the main interpretation of this algorithm should be as subgradient descent.
2. A second issue with the "large enough $\xi$" occurs in the proofs. In the proof of Theorem 1, you rely on choosing  "large enough $\xi$" such that Corollary 1 holds. However, Corollary 1 is a limit statement where you take a limit on $\xi$  after committing to $x^{t-1},y^{t-1}$. On the other hand, Theorem 1 is a statement about choosing  and then generating the sequence of iterates. Thus, I do not think that Corollary 1 can be invoked the way you are doing, since Corollary 1 is only proved for a fixed pair of iterates, whereas you are now having iterates that are determined by your choice of $\xi. More generally, the proof of Theorem 1 is not sufficiently rigorous or detailed. For one, I think you should provide a complete proof rather than say "following the proof of Fasoulakis et al." If you want to invoke Fasoulaskis et al., you should invoke a precise result proven in their paper, and then build from there, not say "the reader can look at their paper and retrace our left-out steps."
3. On forgetfulness experiments: I am not sure these experiments are all that convincing regarding whether FLBR is forgetful or not. First, I couldn't find a specification of what $\delta$ is in these plots. Was it sufficiently small? In either case, ideally you'd try several smaller and smaller $\delta$ choices, to get a sense of whether you chose a small enough $\delta$. Intuitively, it appears plausible that FLBR is forgetful for large $\xi$, due to the "subgradient descent" interpretation that I described above, but these experiments are far too underspecified for me to feel sure either way.

**Questions:**

1. Can the magnitude of $\xi$ be quantified? "Large enough" is too vague in my opinion.
2. Is there any sense in which this is not essentially a result for "subgradient MWU" applied to each player's nonsmooth optimization problem?
3. What were the parameters for the forgetfulness experiments?

---

> ### Author Response · Authors · 2025-11-21
>
> > A missing interpretation in the paper is the following: if we are choosing "large $\xi$" then we are turning the algorithm into a well-known algorithm. In particular, for very large $\xi$, we get that each player is simply best responding to the other player. At that point, you have decoupled the dynamics, and we are simply looking at the intermediate step as a subgradient calculation. At that point, you can instead describe FLBR as follows: we are running subgradient descent, but instead of using projected gradient descent, we use mirror descent with an entropy regularizer. This has implications e.g. for the forgetfulness question. More generally, [...] I think it is fair to say that the main interpretation of this algorithm should be as subgradient descent.
>
> First, we would like to address the above weakness (which also addresses Question 2), on
> whether FLBR leads to decoupled dynamics $\xi$ goes to $\infty$. The way Lemma 2 was stated in our submission, it is true that one could interpret the dynamics as uncoupled. However, Lemma 1 in [Fasoulakis et al.] gives an explicit form on which best response the intermediate steps converge to. In particular we have that the intermediate step converges to:
> $$
> \hat x_i^t = \frac{x_i^{t-1}}{\sum\limits_{j \in BR(y^{t-1})} x_j^{t-1}}
> $$
> where $BR(\cdot)$ stand for the best response polytope. This expression depends on both $x^{t-1}$ and $y^{t-1}$, thus the dynamics remain coupled.
>
> > 2. A second issue with the "large enough " occurs in the proofs. In the proof of Theorem 1, you rely on choosing "$\xi$ large enough " such that Corollary 1 holds. However, Corollary 1 is a limit statement where you take a limit on $\xi$  after committing to $x^{t-1}, y^{t-1}$ . On the other hand, Theorem 1 is a statement about choosing and then generating the sequence of iterates. Thus, I do not think that Corollary 1 can be invoked the way you are doing, since Corollary 1 is only proved for a fixed pair of iterates, whereas you are now having iterates that are determined by your choice of $\xi$. More generally, the proof of Theorem 1 is not sufficiently rigorous or detailed. For one, I think you should provide a complete proof rather than say "following the proof of Fasoulakis et al." If you want to invoke Fasoulaskis et al., you should invoke a precise result proven in their paper, and then build from there, not say "the reader can look at their paper and retrace our left-out steps."
>
> Regarding this weakness and the related question 1, we would like to thank the reviewer again for pointing out this subtle gap in the analysis. Indeed, we cannot invoke the Corollary in the stated way, after having fixed $\xi$ in advance. But there is a straightforward fix. Instead of showing that for every $t$ there exists $\xi$ such that  $$ V(x^{t-1}, y^{t-1}) \leq (\hat{x}^{t})^\top  R{y}^{t-1} - (x^{t-1})^\top  R\hat{y}^t  + \eta$$
> we can prove that there exists $\xi$  such that
> $$ V(x, y) \leq \hat{x}^\top  R{y} - x^\top  R\hat{y}  + \eta. $$
> for every $x, y$ in the respective simplices, where $\hat{x}, \hat{y}$ are the resulting strategies after applying the intermediate step to $x, y$. We have added this in our updated version.
>
> Following the reviewer's comment we also slightly modify our proof to be self-contained (that is why it is slightly more relaxed in the constants, see $7\eta$ instead of $6\eta$), instead of relying on the work of Fasoulakis et al.
>
> > 3. On forgetfulness experiments: I am not sure these experiments are all that convincing regarding whether FLBR is forgetful or not. First, I couldn't find a specification of what $\delta$ is in these plots. Was it sufficiently small? In either case, ideally you'd try several smaller and smaller $\delta$ choices, to get a sense of whether you chose a small enough $\delta$ . Intuitively, it appears plausible that FLBR is forgetful for large $\xi$, due to the "subgradient descent" interpretation that I described above, but these experiments are far too underspecified for me to feel sure either way.
> What were the parameters for the forgetfulness experiments?
>
>
> The reviewer is right that we omitted to mention our choice of $\delta$ and we correct our oversight in the updated version. We picked $\delta = 10^{-2}$, same as (Cai et al.). If after our answer in question 1 the reviewer agrees that it is not subgradient descent, perhaps they can see that the choice of $\delta$ does not matter as much. Still, if the reviewer wishes we can incorporate more plots with smaller choices of $\delta$ in an additional appendix. At a high level, the difference that smaller $\delta$s make is that a higher number of iterations is required among both methods before the phase of start convergence starts.

---

> > ### Comment · Reviewer_rnbp · 2025-11-26
> >
> > I do not understand your claim about which best response we are converging to. I think there might be some indexing errors in the expression you gave?
> > In any case though, you state yourself in Lemma 2 that as $\xi$ goes to infinity, we get that the intermediate step converges to a best response. If that is so, then this is mirror descent with the entropy DGF applied to the non-smooth representation of each player's problem, as I had stated in my review. Are you saying that Lemma 2 is wrong, or am I misunderstanding what you were trying to say?

---

> > > ### Author Response · Authors · 2025-11-27
> > >
> > > Thank you for your response and we apologize for any confusion caused. Maybe we have indeed misunderstood each other, so let us clarify our point.
> > > First of all, just to be sure about what you asked us: when you refer to decoupled dynamics in your review, we assume that you mean that each player is able to keep computing her own update to her current strategy without ever the need to see what the other player's strategy is, right?
> > >
> > > Given this, in your review you are suggesting that when $\xi\rightarrow\infty$ the dynamics can be decoupled because the row player at any time $t$ could compute herself a best response to her previous strategy $x^{t-1}$ (this would be the $\hat y^t$), and then based on that she can update her own strategy to $x^t$, according to the update step of FLBR.
> > >
> > > What we are saying is that when $\xi\rightarrow\infty$, the intermediate strategies $\hat{x}, \hat{y}$ do not just converge to an arbitrary best response, but to a particular one, namely, to the equation written in our rebuttal. I.e.,
> > > Lemma 1 in [Fasoulakis et al.] states that for every $i\in BR(y^{t-1})$, $\hat x_i^t$ converges to:
> > > $$
> > > \hat x_i^t = \frac{x_i^{t-1}}{\sum\limits_{j \in BR(y^{t-1})} x_j^{t-1}}
> > > $$
> > > On the other hand, for $i\not\in BR(y^{t-1})$, $\hat x_i^t$ converges to 0.
> > >
> > > Therefore the intermediate step converges to a mixed strategy that randomizes over pure best response strategies to $y^{t-1}$. Hence, we need to know $y^{t-1}$ in order to compute the form of $\hat x^t$, and this is why we think the dynamics are not decoupled when $\xi\rightarrow\infty$.

---

### Official Review · Reviewer_i1TU · 2025-10-26

**Soundness:** 3
**Presentation:** 2
**Contribution:** 2
**Rating:** 4
**Confidence:** 3

**Summary:**

This paper studies the Forward Looking Best Response Dynamics (FLBR) and investigates several properties of the dynamics in two player zero sum games. In particular, the paper explores the last iterate convergence rate (to both approximate and exact NE), showing that a geometric rate dependent on game dimension can be obtained for approximate NE, while the convergence rate to exact (unique) NE depends on the Jacobian of the game. The paper also shows that FLBR is not no-regret, and is forgetful in the sense of Cai et al 2024. Finally, some empirical evidence is shown that FLBR performs comparatively or even better than OGDA, though this seems to be a game-class dependent behavior.

**Strengths:**

- The paper has a clear structure and the research problems investigated are reasonable. This results in a stronger understanding of the FLBR dynamics in comparison to other 'standard' accelerated methods.
- The use of ideas from information theory in the proof of Theorem 2 is interesting and could be a useful tool moving forward.

**Weaknesses:**

- The main selling point of FLBR compared to OGDA/OMWU is the geometric rate of convergence to approximate NE, as opposed to a $O(1/\sqrt{t})$ for OGDA for example. However, this is very much dependent on the approximation required of the NE, as once the game dependent geometric rate kicks in, the separation seems to disappear and OGDA seems to practically still perform better (except in 'structured' games). I think this reduces the significance of the results somewhat, as there is no clear reason to implement FLBR unless you know the game is 'structured' in some way. One way to address this would be to theoretically justify some classes of games (e.g. symmetric) for which the game dependent constant is small, thereby explaining the speedup of FLBR compared to OGDA.
- The discussion on forgetfulness is interesting but not convincing to me. The example used in Cai et al 2024 was a degenerate example for the case of OFTRL, but while the paper states that the larger intermediate update avoids forgetfulness, this statement is not substantiated. The Cai paper shows negative (non-forgetful) results, but to prove that FLBR is forgetful would require more work. For example consider the setting of Braverman et al (2018) and Kumar et al (2024), both cited in Cai's paper, who studied a similar property to forgetfulness. Kumar et al proved that OGDA is not exploitable by an adversary, and is thus in some sense 'forgetful'. However this is in the context of no-regret algorithms, which FLBR is not. I believe there is more depth and subtlety here to be explored, and would be very curious to know if the authors have studied this in more detail as it would improve the paper.
- While the paper is quite clear in it's structure, I found the writing to overall be very stylized and occasionally imprecise, which reduces the clarity of exposition. For instance, in the last paragraph of the introduction, the authors describe 'positive' and 'negative' results without clarifying what they are, ending with the vague statement that 'the landscape is overall less clear'. In the experimental section, FLBR is described to perform 'close enough' to OGDA in random gaussian games, and 'FLBR comes on top/OMWU is far away' in RPS games. The imprecise language makes it unclear what the threshold for 'close enough' should be.

In light of the above, I find the paper is marginally under the threshold of acceptance to ICLR, as the writing needs more polishing and more theoretical justification is needed for FLBR compared to existing methods.

**Questions:**

- Purely out of curiosity, how does FLBR perform in other game classes which are non-zero-sum? For example, I would expect FLBR to converge quickly in potential games, but does it also outperform OGDA/OMWU in this class of games?

---

> ### Author Response · Authors · 2025-11-21
>
> > The main selling point of FLBR compared to OGDA/OMWU is the geometric rate of convergence to approximate NE, as opposed to a $O(1/\sqrt{t})$ for OGDA for example. However, this is very much dependent on the approximation required of the NE, as once the game dependent geometric rate kicks in, the separation seems to disappear and OGDA seems to practically still perform better (except in 'structured' games). I think this reduces the significance of the results somewhat, as there is no clear reason to implement FLBR unless you know the game is 'structured' in some way. One way to address this would be to theoretically justify some classes of games (e.g. symmetric) for which the game dependent constant is small, thereby explaining the speedup of FLBR compared to OGDA.
>
> We certainly agree that establishing small game-dependent constants for some classes of games under FLBR is a very interesting direction. At the moment we do not have any theoretical results of this form.
>
> > The discussion on forgetfulness is interesting but not convincing to me. The example used in Cai et al 2024 was a degenerate example for the case of OFTRL, but while the paper states that the larger intermediate update avoids forgetfulness, this statement is not substantiated. The Cai paper shows negative (non-forgetful) results, but to prove that FLBR is forgetful would require more work. For example consider the setting of Braverman et al (2018) and Kumar et al (2024), both cited in Cai's paper, who studied a similar property to forgetfulness. Kumar et al proved that OGDA is not exploitable by an adversary, and is thus in some sense 'forgetful'. However this is in the context of no-regret algorithms, which FLBR is not. I believe there is more depth and subtlety here to be explored, and would be very curious to know if the authors have studied this in more detail as it would improve the paper.
>
> We agree that this section may not be conclusive enough. We stress that this was not meant to serve as part of our most important results. But still for the sake of completeness, and in light of the recent paper on forgetfulness, we felt like this should be included
> in order to have a thorough study of the algorithm’s properties.
>
> > While the paper is quite clear in it's structure, I found the writing to overall be very stylized and occasionally imprecise, which reduces the clarity of exposition. For instance, in the last paragraph of the introduction, the authors describe 'positive' and 'negative' results without clarifying what they are, ending with the vague statement that 'the landscape is overall less clear'. In the experimental section, FLBR is described to perform 'close enough' to OGDA in random gaussian games, and 'FLBR comes on top/OMWU is far away' in RPS games. The imprecise language makes it unclear what the threshold for 'close enough' should be.
>
> Any statements that we made on the experimental comparisons are derived by our presented plots, and one can see in the plots the more precise differences between the methods. In the updated version, we tried to write more explicitly the outcomes of the comparisons, on the performance of FLBR vs OGDA.
>
> > Purely out of curiosity, how does FLBR perform in other game classes which are non-zero-sum? For example, I would expect FLBR to converge quickly in potential games, but does it also outperform OGDA/OMWU in this class of games?
>
> We had not considered this direction for research and we thank the reviewer for pointing it out. We have expanded Appendix D.3 in the updated version to include some preliminary experiments with potential games. As the reviewer suspected, FLBR does converge.

---

> > ### Comment · Reviewer_i1TU · 2025-11-28
> > **Response to rebuttal**
> >
> > Thank you for your response. After reading the other reviewers' comments, it seems there is an agreement that the theoretical guarantees provided by the present version are not particularly convincing. Nevertheless, I think that with some additional analysis, the paper would be strong, so I will maintain my score.

---

### Official Review · Reviewer_FNqj · 2025-10-28

**Soundness:** 2
**Presentation:** 3
**Contribution:** 2
**Rating:** 4
**Confidence:** 3

**Summary:**

This paper studied the dynamical behaviors of Forward Looking Best Response MWU (FLBR-MWU), which was first proposed by Fasoulakis et al. (2022), in two-player zero-sum games. The main results include:
* An analysis of the convergence rate of FLBR-MWU. Specifically, the behavior of the algorithm can be divided into two phases: First, it converges to an approximate equilibrium at an inverse exponential rate. Second, by analyzing the Jacobian of the dynamics at equilibrium, the authors show that the dynamics also exhibit an inverse exponential rate when the strategy is sufficiently close to the equilibrium.
* The authors provide examples to show that FLBR-MWU is indeed not a no-regret algorithm.
* Experiments on the forgetful and convergence rate behaviors are provided for FLBR-MWU, which indicate that FLBR-MWU is comparable with other popular algorithms such as Optimistic Gradient Descent-Ascent.

**Strengths:**

* The theoretical results enchance our understanding of the convergence behaviors of FLBR-MWU algorithms.
* Numerical experiments provide intuitions on how the performance of FLBR-MWU compared with several other popular algorithms like Optimistic Gradient Descent-Ascent and Optimistic MWU.

**Weaknesses:**

* The theoretical results, which state that FLBR-MWU first converges to an approximate equilibrium at an inverse exponential rate depending only on the game dimension (i.e., Theorem 1), represent a slightly modified version of the theorem in (Fasoulakis et al., 2022) in terms of the duality gap. Moreover, the Jacobian-type analysis of dynamics near an equilibrium in Theorem 2 is also well-known in the literature; for example, see Theorem 4 in (Fasoulakis et al., 2022) and Section 3 in (Daskalakis & Panageas, 2018). These types of Jacobian analyses usually provide an exponential convergence rate related to the eigenvalues of the Jacobian at equilibrium. Comparing the results of Theorem 2 with these related works, it is unclear what new information is provided. This makes the theoretical contribution incremental.

* Given that Extra Gradient methods are not no-regret algorithms, and the FLBR-MWU algorithm is very similar to the extragradient method, with the only difference being the use of a different step size in the intermediate step (Mertikopoulos et al., 2018), it is not surprising that FLBR-MWU is also not a no-regret algorithm.

* The discussion of the forgetfulness property of FLBR-MWU (Section 4.2) remains at a rough level, with only one toy $2 \times 2$ example provided to explain the phenomenon. The discussion is very high-level and makes it difficult for readers unfamiliar with the work of (Cai et al., 2024) to grasp the key points. For example, it is stated that *"if a method is not forgetful, the produced strategies can get stuck at almost the same profile over many iterations, which slows down convergence."* However, in Figure 1, it is not explained how the algorithms get stuck at nearly the same profile. I suggest that the authors further clarify this phenomenon by providing a comparison with algorithm trajectories that lack the forgetfulness property.

Reference:

Mertikopolous et al., Optimistic mirror descent in saddle-point problems: Going the extra (gradient) mile

Daskalakis & Panageas, The Limit Points of (Optimistic) Gradient Descent in Min-Max Optimization

**Questions:**

1. Compared with the Jacobian-type analysis appearing in related works such as (Daskalakis & Panageas, 2018; Fasoulakis et al. 2022)), which provides an exponential convergence rate in terms of the eigenvalues of the Jacobian, what new information does Theorem 2 provide?

The authors claimed

> *"in order to obtain a rate of convergence, we give a more refined analysis, based on a technique utilized in Nakagawa et al. (2021)"*
(lines 298–299).

Could you provide more explanation on how the technique in Nakagawa et al. (2021) can provide new insights into the local convergence behavior of the algorithms that the standard Jacobian-type analysis in (Daskalakis & Panageas, 2018; Fasoulakis et al., 2022) cannot provide?

---

2. Compare with Optimistic MWU and Optimistic GDA (Wei et al., 2021), what is the benifits of FLBR-MWU in terms of convergence rate or other aspects?

In the paper, the authors claimed that

> *"We view as advantages of our analysis that it yields a simpler and more intuitive proof compared to Wei et al. (2021), and it also establishes fast (non-game-dependent) convergence to an approximate equilibrium before approaching the exact solution"* (lines 85–88).

Can the authors provide more explanation on this point? Specifically, what are the game-dependent constants that appear in the results of Wei et al. (2021)?

Moreover, in Section 4.2 of the current work, the authors state that Optimistic GDA has a better convergence rate than OMWU, and from the experiments it can be observed that Optimistic GDA performs even better than FLBR-MWU. Thus, I wonder whether the existing convergence rate bounds of Optimistic GDA also suffer from the issue of game-dependent constants?

---

3. As the authors stated in Section 4.2, Optimistic-GDA has better performance than Optimistic-MWU. I'd like to know whether it is possible that FLBR-GDA could also perform better than the FLBR-MWU algorithm proposed in the current work.

---

> ### Author Response · Authors · 2025-11-21
>
> > 1. Compared with the Jacobian-type analysis appearing in related works such as (Daskalakis & Panageas, 2018; Fasoulakis et al. 2022)), which provides an exponential convergence rate in terms of the eigenvalues of the Jacobian, what new information does Theorem 2 provide?
>
> The paper by Fasoulakis et al. 22, does not provide any concrete analysis on the convergence rate. It
> is proved there that the eigenvalues are less than 1, establishing asymptotic convergence, but without
> stating any formula for the dependence of the rate on the eigenvalues. It is generally perceived that
> if the eigenvalues are less than one, then this implies geometric convergence but we have not found
> any specific reference with an explicit formula for the duality gap, that is directly applicable to our
> setting. Our paper provides explicitly how the rate depends on the maximum eigenvalue, and this is
> where the technique of the Nakagawa et al. 2021 paper is used to pinpoint this dependence. Finally,
> regarding the analysis in the (Daskalakis Panageas, 2018), this concerns OGDA, and therefore it
> does not seem to imply anything (at least not immediately) for the dynamics we study here.
>
> > Given that Extra Gradient methods are not no-regret algorithms, and the FLBR-MWU algorithm is very similar to the extragradient method, with the only difference being the use of a different step size in the intermediate step (Mertikopoulos et al., 2018), it is not surprising that FLBR-MWU is also not a no-regret algorithm.
> The discussion of the forgetfulness property of FLBR-MWU (Section 4.2) remains at a rough level, with only one toy  example provided to explain the phenomenon. The discussion is very high-level and makes it difficult for readers unfamiliar with the work of (Cai et al., 2024) to grasp the key points. For example, it is stated that "if a method is not forgetful, the produced strategies can get stuck at almost the same profile over many iterations, which slows down convergence." However, in Figure 1, it is not explained how the algorithms get stuck at nearly the same profile. I suggest that the authors further clarify this phenomenon by providing a comparison with algorithm trajectories that lack the forgetfulness property.
>
> We agree that FLBR being regretful is not surprising but we feel like this should be included in order to have a thorough study of the algorithm's properties. This was not meant to serve as part of our most important results, but still the lack of regret or not was left unanswered  by Fasoulakis et al. 2022. Our reasoning is the same for the more recent forgetfulness property, which we acknowledge that a reader may not be as familiar with and we therefore have slightly improved the exposition in our new uploaded version. If the reviewer feels that the section is still unclear, we can add an additional appendix to expand upon it.
>
> > 2. Compare with Optimistic MWU and Optimistic GDA (Wei et al., 2021), what is the benifits of FLBR-MWU in terms of convergence rate or other aspects?
> Can the authors provide more explanation on this point? Specifically, what are the game-dependent constants that appear in the results of Wei et al. (2021)?
> Moreover, in Section 4.2 of the current work, the authors state that Optimistic GDA has a better convergence rate than OMWU, and from the experiments it can be observed that Optimistic GDA performs even better than FLBR-MWU. Thus, I wonder whether the existing convergence rate bounds of Optimistic GDA also suffer from the issue of game-dependent constants?
>
> Regarding the game convergence constants for OMWU in [Wei et al.], in the proof of their Theorem 2 (Appendix D in their full paper) they conclude with an expression that depends on $C_2$ in the base and on $T_0$ in the exponent. By following their definitions, both parameters depend on $\xi$ (see their Definition 2) and $\epsilon$ (see their Definition 4). Both those quantities depend on the game, and especially $\epsilon$ creates an exponential dependency. Hence both the initial phase of slow convergence $T_0$ can be exponentially large in the entries of the payoff matrix and the same holds for the base of exponential rate; that is it can be written as $c^{t-T_o}$ where $1-c$ is exponentially small.
> To conclude: the game-dependent constants for OMWU are incomparable to our constants. But our main advantage is that our analysis yields a much simpler and more direct proof, much easier to follow its steps, and with the clear advantage that till we get to an approximate equilibrium we do not have any game-dependent constants.
>
> Related to this, and regarding the question of the reviewer on whether the existing convergence rate of Optimistic GDA also suffers from the issue of game-dependent constants: yes, the reviewer is correct. For OGDA, there are known convergence rates for $\varepsilon$-NE of the form $k \log(\frac{1}{\varepsilon})$ where $k$ depends on the matrix entries (and it usually is some condition number of the payoff matrix).

---

> > ### Author Response · Authors · 2025-11-21
> >
> > > 3. As the authors stated in Section 4.2, Optimistic-GDA has better performance than Optimistic-MWU. I'd like to know whether it is possible that FLBR-GDA could also perform better than the FLBR-MWU algorithm proposed in the current work.
> >
> > We have not worked on this so far, but it is a very interesting point for future work. Our first thought was that it should not work because while a large $\xi$ will maintain the intermediate iterates inside the simplex in an MWU step, this would not be the case in GDA. On the other hand, if we view them both as variants of OMD, it is natural to expect the idea of "looking forward" to adapt, maybe via a suitable projection.

---

### Official Review · Reviewer_j6dJ · 2025-10-31

**Soundness:** 3
**Presentation:** 3
**Contribution:** 2
**Rating:** 6
**Confidence:** 3

**Summary:**

his paper investigates the last-iterate convergence properties of an algorithm called Forward Looking Best-Response Multiplicative Weights Update (FLBR-MWU) in two-player bilinear zero-sum games. FLBR-MWU is a variant of the Multiplicative Weights Update (MWU) method inspired by the extragradient idea, characterized by using a larger learning rate ξ in its intermediate prediction step and a smaller learning rate η in the final update step. The authors establish, for the first time, explicit convergence rates for this algorithm:

- Prior to reaching a approximate Nash equilibrium, the duality gap converges at a geometric rate independent of the game matrix.
- If the game admits a unique Nash equilibrium, once sufficiently close to equilibrium, the duality gap continues to converge to zero at a geometric rate dependent on the spectral radius of the Jacobian matrix.

Additionally, the authors prove that FLBR is not a no-regret algorithm and demonstrate experimentally that it performs comparably to or better than Optimistic Gradient Descent Ascent (OGDA) in normal-form games (NFGs). The paper also notes that FLBR exhibits a "forgetfulness" property, which may contribute to accelerated convergence.

**Strengths:**

1. **Clear theoretical contribution**: The paper provides the first explicit last-iterate geometric convergence rates for FLBR-MWU, resolving an open question posed by Fasoulakis et al. (2022).
2. **Well-structured two-phase analysis**: The convergence process is cleanly divided into two phases—(i) rapid approximation toward an approximate equilibrium with a game-matrix-independent rate, and (ii) refined convergence to the exact equilibrium with a rate dependent on the Jacobian’s spectral radius but offering greater precision.
3. **Elegant and intuitive proof**: Compared to the KL-divergence-based analysis of OMWU in Wei et al. (2021), the current paper’s analysis of the duality gap is more direct and cleverly adapts convergence techniques from the Arimoto–Blahut algorithm in information theory.

**Weaknesses:**

1. **Incomplete coverage of related work**: While discussing last-iterate convergence methods, the paper focuses exclusively on optimistic/extragradient-type approaches (e.g., OGDA, EG) and overlooks other relevant directions, such as:
   - Regularization-based methods [1, 2];
   - Negative momentum-based methods [3].
     These approaches have demonstrated effectiveness not only in bilinear games but also in more general settings like extensive-form games (EFGs). Including them in the Related Work section would better contextualize the paper’s contribution.
2. **Strong theoretical assumptions**: The geometric convergence in the second phase relies on the assumption of a unique Nash equilibrium. Although the authors note that this condition holds "almost always" in a measure-theoretic sense, it may limit the practical applicability of the result. Nevertheless, I do not believe this significantly diminishes the paper’s overall contribution.
3. **Typo**: For instance, an extraneous comma appears in line 148 (“w.r.t.,”).

[1] Sokota, S., et al.; A Unified Approach to Reinforcement Learning, Quantal Response Equilibria, and Two-Player Zero-Sum Games.

[2] Liu, M., et al.; The Power of Regularization in Solving Extensive-Form Games.

[3] Fang, Z., et al.; Rapid Learning in Constrained Minimax Games with Negative Momentum.

**Questions:**

**Generalization capability**: In Appendix D.3, the paper attempts to apply FLBR to non-bilinear convex-concave problems but observes non-convergence. Could the algorithm be adapted—e.g., by incorporating regularization or negative momentum—to achieve convergence in broader classes of min-max problems?

---

> ### Author Response · Authors · 2025-11-21
>
> >1.  Incomplete coverage of related work: While discussing last-iterate convergence methods, the paper focuses exclusively on optimistic/extragradient-type approaches (e.g., OGDA, EG) and overlooks other relevant directions, such as:
> Regularization-based methods [1, 2];
> Negative momentum-based methods [3]. These approaches have demonstrated effectiveness not only in bilinear games but also in more general settings like extensive-form games (EFGs). Including them in the Related Work section would better contextualize the paper’s contribution.
>
> Indeed, we focused exclusively on optimistic/extra-gradient approaches. Now, we have updated our manuscript to improve the coverage of the suggested related works.
>
> > 2. Strong theoretical assumptions: The geometric convergence in the second phase relies on the assumption of a unique Nash equilibrium. Although the authors note that this condition holds "almost always" in a measure-theoretic sense, it may limit the practical applicability of the result. Nevertheless, I do not believe this significantly diminishes the paper’s overall contribution.
>
> We share the reviewer's view that this is a minor caveat and we would also like to add that we expect that it can be dropped. Our intuition is the following: given that the convergence to an approximate equilibrium holds without assuming uniqueness, we can get close enough to an actual equilibrium so that the local contraction property holds even in the presence of other equilibria, which are far enough to not effect us. Our conjecture therefore is that in the absence of uniqueness, we can have convergence to the (convex) set of equilibria. We have also observed experimentally that the algorithm converges in the presence of multiple equilibria.
>
> > 3. Typo: For instance, an extraneous comma appears in line 148 (“w.r.t.,”).
>
> Typos and other writing issues pointing by the other reviewers have beed addressed in our revision.
>
> > Generalization capability: In Appendix D.3, the paper attempts to apply FLBR to non-bilinear convex-concave problems but observes non-convergence. Could the algorithm be adapted—e.g., by incorporating regularization or negative momentum—to achieve convergence in broader classes of min-max problems?
>
> We have included the experiments in the appendix with the method as is, to provide more comparisons with OMWU and OGDA, which are known to maintain convergence in the bilinear setting. We think that FLBR can be adapted to provide guarantees
> in a broader setting and consider this as interesting topic for future work.

---

### Author Response · Authors · 2025-11-21

We thank all the reviewers for reading our work and providing valuable feedback, which we have incorporated in the updated manuscript. We also respond to each reviewer separately via the corresponding comments.

---

### Meta-Review · Area_Chair_aug4 · 2026-01-12

**Summary:**

The paper focuses on the analysis of a variant of Multiplicative Weights Update, called Forward Looking Best Response MWU applied to two-player zero-sum games. It was established in 2022 that Forward Looking Best Response MWU exhibits last iterate convergence to Nash equilibrium in two-player zero-sum games and left open the question about the rate of convergence. This paper established exponential rate of convergence $c^t$ for the duality gap (duality gap $\epsilon$ gives $\epsilon$-NE).

**Reviewer Concerns:**

There were a few concerns about the novelty of the analysis of the paper, the spectral analysis of the Jacobian in particular and about the related work. A few concerns also were raised about the write-up and details on the proofs. Overall the AC believes this is a nice theoretical contribution, however the AC cannot recommend acceptance based on the reviewers' scores.

**Reviewer Scores:**

The AC does not believe that the scores would be substantially changed and as a result the paper is below the bar for acceptance.

---

### Decision · Program_Chairs · 2026-01-26

Reject